

# Component-oriented acausal modeling of the dynamical systems in Python language on the example of the model of the sucker rod string

Volodymyr B. Kopei, Oleh R. Onysko and Vitalii G. Panchuk

Department of Computerized Mechanical Engineering, Ivano-Frankivsk National Technical University of Oil and Gas, Ivano-Frankivsk, Ukraine

Corresponding author
Volodymyr B. Kopei,
volodymyr.kopey@nung.edu.ua

## ABSTRACT

Typically, component-oriented acasual hybrid modeling of complex dynamic systems is implemented by specialized modeling languages. A well-known example is the Modelica language. The specialized nature, complexity of implementation and learning of such languages somewhat limits their development and wide use by developers who know only general-purpose languages. The paper suggests the principle of developing simple to understand and modify Modelica-like system based on the general-purpose programming language Python. The principle consists in: (1) Python classes are used to describe components and their systems, (2) declarative symbolic tools SymPy are used to describe components behavior by difference or differential equations, (3) the solution procedure uses a function initially created using the SymPy lambdify function and computes unknown values in the current step using known values from the previous step, (4) Python imperative constructs are used for simple events handling, (5) external solvers of differential-algebraic equations can optionally be applied via the Assimulo interface, (6) SymPy package allows to arbitrarily manipulate model equations, generate code and solve some equations symbolically. The basic set of mechanical components (1D translational "mass", "spring-damper" and "force") is developed. The models of a sucker rods string are developed and simulated using these components. The comparison of results of the sucker rod string simulations with practical dynamometer cards and Modelica results verify the adequacy of the models. The proposed approach simplifies the understanding of the system, its modification and improvement, adaptation for other purposes, makes it available to a much larger community, simplifies integration into third-party software.

# INTRODUCTION

As is known, component-oriented simulation modeling is based on the separation of a complex system model into simple components. The component describes the mathematical model of the corresponding physical object (mass, spring, electrical

resistance, hydraulic resistance, hydraulic motor, etc.), which is formulated as an algebraic, differential or difference equation. The components are connected through ports (pins, flanges), which define a set of variables for the component interaction (*Elmqvist, 1978*; *Fritzson, 2015*). The components and ports are stored in software libraries. Usually, it is possible to develop new components. The multi-domain modeling allows using together of components that differ in physical nature (mechanical, hydraulic, electric, etc.). The component-oriented modeling can be based on causal modeling or acausal modeling (*Fritzson, 2015*). In the first case, the component receives the $x$ signal at the input, performs a certain mathematical operation $f(x)$ on it and returns the $y$ result to the output. In this case, the modeling is realized by imperative programming by assigning the value of the $f(x)$ expression to the $y$ variable. In the second case, the signal of the connected components can be transmitted in two directions. Such modeling is realized by declarative programming by solving the equation $y = f(x)$, where the unknown can be $x$ or $y$. Here, the variables $x$ and $y$ are some physical quantities, and the equation $y = f(x)$ is the physical law that describes their relationship. It allows us to simplify the creation of the model, to focus on the physical formulation of the problem, but not on the algorithm for solving it. It is also possible to avoid errors that are typical for imperative programming.

The behavior of these models is most often described by a system of ordinary differential equations (ODEs) or a differential-algebraic system of equations (DAEs), which are solved by the finite difference method—a numerical method based on the replacement of differential operators by difference schemes. As a result, the system of differential equations is replaced by the system of algebraic equations.

The solution of non-stationary problems by the finite difference method is the iterative process—there is a solution of the stationary problem for the given time point at each iteration. Explicit and implicit difference schemes are used for this purpose. Explicit schemes immediately find unknown values, using information from the previous iterations. Using the implicit scheme requires the solution of a difference equation because unknown values can be in the right and left sides of the equation. The explicit Euler difference scheme is simple to implement, but it often has numerical instability and low accuracy. The analysis of the Euler method is described in detail in (*Atkinson, 1989*). To improve accuracy and stability it is desirable to apply modified Euler methods, such as the Runge–Kutta method (*Runge, 1895*).

For simulations of complex dynamic multi-domain systems such specialized equation-based modeling languages are developed: Dymola (*Elmqvist, 1978*), APMonitor (*Hedengren et al., 2014*), ASCEND (*Piela, McKelvey & Westerberg, 1993*), gPROMS (*Barton & Pantelides, 1993*), Modelica (*Fritzson & Engelson, 1998*), MKL, Modelyze (*Broman, 2010*). Among them, Modelica is the most popular free language for component-oriented modeling of such systems. Its main features: free, object-oriented, declarative, focused on hybrid (continuous and discrete) component-oriented modeling of complex multi-domain physical systems, it supports the construction of hierarchical models, is adapted for visual programming and widely used for research in various fields (*Fritzson, 2015*). Free Modelica Standard Library has about 1,280 components. There are free and

commercial simulation environments for Modelica—OpenModelica, JModelica.org, Wolfram SystemModeler, SimulationX, MapleSim, Dymola, LMS Imagine.Lab AMESim.

The known problem of specialized languages is the complexity of the modifications and improvements and a relatively small community of developers. They are not very well suited for experimenting with evolutions of modeling capabilities (*Elmqvist, Henningsson & Otter, 2016*). In particular, developers of some languages have encountered the problem of variable structure systems modeling where the structure and number of equations can change at run-time (*Fritzson, Broman & Cellier, 2009*; *Nikolić, 2016*). Some problems can be solved by using interfaces to general-purpose languages (*Åkesson et al., 2010*; *Hedengren et al., 2014*). But it is usually more difficult to learn a new language than to learn a component or a library of a familiar programming language.

As a rule, general-purpose languages, in comparison with specialized languages, are more widespread, easy to learn thanks to typical imperative and object-oriented constructs, have wider applicability, better interoperability with the third party software and a large number of heterogeneous packages. Therefore, the mentioned problems are less common in modeling systems that are based on general-purpose programming languages: PyDSTool (*Clewley et al., 2007*)—Python-based Dynamical Systems Toolkit with support for symbolic manipulation, hierarchical structures and hybrid models; Ariadne—a C++ library for formal verification of cyber-physical systems, using reachability analysis for nonlinear hybrid automata (*Benvenuti et al., 2014*); Assimulo—Python-package that combines a variety of different ODE/DAE solvers via a common high-level interface (*Andersson, Führer & Åkesson, 2015*); DAE Tools—equation-based object-oriented modeling, simulation and optimization software with hybrid approach (*Nikolić, 2016*); Modia.jl—Modelica-like language that is directly defined and implemented with Julia's meta-programming constructs and is designed tightly together with the symbolic and numeric algorithms (*Elmqvist, Henningsson & Otter, 2016*); SimuPy—a Python framework for simulating interconnected dynamical system models (*Margolis, 2017*); Sims.jl—a Julia package for equation-based hybrid modeling and simulations, similar to Modelyze (*Short, 2017*); GEKKO—a Python package for machine learning and optimization of mixed-integer and differential algebraic equations (*Beal et al., 2018*). However, most of these systems either have a complex code that is difficult to understand and modify (e.g., have their own symbolic processors), or use state-of-the-art ODE/DAE solvers that are implemented in low-level languages, which rarely allow modification to an untrained user. The implementation, modification and improvement of such systems can be simplified if the difference equations are used to describe the model instead of differential equations. In addition, difference equations are also often used to model dynamical systems. In general, the modeling system should allow various types of equations.

The advantages of modeling systems based on general-purpose programming languages are described in detail in papers (*Nikolić, 2016*; *Elmqvist, Henningsson & Otter, 2016*). Python language (*Van Rossum & Drake, 1995*) is a good choice mainly due to its features: multi-paradigm, object-oriented, intuitive with code readability and improved

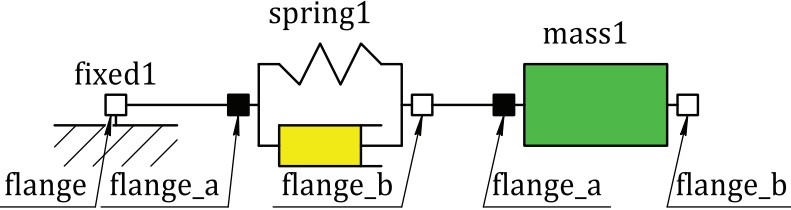

**Figure 1 Component diagram of the oscillator model.**

programmer's productivity, highly extensible, portable, open-source, large community and extensive libraries as mathematical libraries SymPy and SciPy. SymPy is a Python library for symbolic mathematics (*Meurer et al., 2017*). SciPy is a fundamental library for scientific computing (*Oliphant, 2007*).

This work suggests the principle of developing simple to understand and modify Modelica-like system based on the general-purpose programming language Python. The principle consists in: (1) Python classes are used to describe components and their systems, (2) declarative symbolic tools SymPy are used to describe components behavior by difference or differential equations, (3) the solution procedure uses a function initially created using SymPy `lambdify` and computes unknown values in the current step using the known values from the previous step, (4) Python imperative constructs are used for simple events handling, (5) external DAEs solvers can optionally be applied via the Assimulo interface, (6) SymPy package allows to arbitrarily manipulate model equations, generate code and solve some equations symbolically. The principle of the system is described by examples of models of a sucker rod string that is used in the oil industry to connect surface and downhole components of a rod pumping system.

## METHODS AND IMPLEMENTATION

### Description of modeling principles in Modelica language

First, we describe the modeling principles in Modelica using an example of a simple mechanical translational oscillator. The oscillator consists of such components as `Mass`, `SpringDamper` and `Fixed` (Fig. 1). The `SpringDamper` component is designed to simulate the elastic-damper properties of the damped oscillator. The `Mass` component simulates the inertial properties of the oscillator. The `Fixed` component simulates the fixed point of the oscillator. The module code that describes this model is explained below (Listing S1). In order to simplify the model, these classes differ slightly from the corresponding classes of the standard Modelica library (*Fritzson, 2015*).

A class in Modelica describes the set of similar objects (components). The `Flange` class describes the concept of a mechanical flange. Its real-type `s` variable corresponds to the absolute position of the flange. Its value should be equal to the value of the `s` variables of other flanges connected to this flange. The real-type `f` variable corresponds to the force on the flange. It is marked by the `flow` keyword, which means that the sum of all forces at the connection point is zero.

```
connector Flange // class-connector
  Real s; // variable (positions at the flange are equal)
  flow Real f; // variable (sum of forces at the flange is zero)
end Flange;
```

The `Fixed` class describes the concept of a fixed component with one flange, for example, `fixed1` (Fig. 1). It has the real-type `s0` variable, which corresponds to the absolute position of the flange, and the `flange` object of the `Flange` class, designed to connect this component to others. The `s0` variable is marked by the `parameter` keyword, which means that it can be changed only at the start of the simulation. After the `equation` keyword, an equation describing the behavior of this component is declared—the `flange` object position must be equal to the `s0` value.

```
model Fixed // class-model
  parameter Real s0=0; // parameter (constant in time)
  Flange flange; // object of class Flange
equation // model equations
  flange.s = s0;
end Fixed;
```

The `Transl` class describes an abstract component that has two flanges—`flange_a` and `flange_b`. It is the base class for mechanical translational components with two flanges.

```
partial model Transl // class-model
  Flange flange_a; // object of class Flange
  Flange flange_b; // object of class Flange
end Transl;
```

The Mass class inherits the `Transl` class and describes the sliding mass with inertia. The example of such component is `mass1` (Fig. 1). The `extends Transl` command means inheriting members of the `Transl` class in such a way that they become members of the `Mass` class. That is, the `Mass` component will also have two flanges (`flange_a` and `flange_b`). In addition, this class has the `m` parameter (mass) and the variables: `s` (position), `v` (speed), `a` (acceleration). The expression `start=0` is the default initial condition. After the `equation` keyword the system of the differential and algebraic equations, which describes the behavior of this component, is given. The `der` keyword means the derivative with respect to time $t$ ($v = \mathrm{d}s/\mathrm{d}t$, $a = \mathrm{d}v/\mathrm{d}t$).

```
model Mass // class-model
  extends Transl; // inheritance of class Transl
  parameter Real m(min=0, start=1); // parameter
  Real s; // variable
  Real v(start=0); // variable with initial condition
  Real a(start=0); // variable with initial condition
equation // model equations
  m*a = flange_a.f + flange_b.f;
```

```
    a = der(v);
    v = der(s);
    flange_a.s = s;
    flange_b.s = s;
  end Mass;
```

The `SpringDamper` class inherits the `Transl` class and describes the linear 1D translational spring and damper in parallel (Listing S1). The example of such component is `springDamper1` (Fig. 1). The class has the parameters `c` (spring constant), `d` (damping constant) and the variables `s_rel` (relative position), `v_rel` (relative speed), `f` (force at `flange_b`). After the `equation` keyword the system of differential-algebraic equations of this component is given.

The `Oscillator` class describes the spring-mass system (Fig. 1). It contains three components `mass1`, `spring1`, `fixed1`, which are described by the classes `Mass`, `SpringDamper` and `Fixed`, respectively. The values of parameters and initial conditions of these components are shown in round brackets.

```
model Oscillator // class-model
  Mass mass1(s(start=-1), v(start=0), m=3961.0); // object with
initial conditions
  SpringDamper spring1(c=44650.0, d=2120.7); // object
  Fixed fixed1(s0=0); // object
equation // additional equations
  // creates a system of equations (see Flange class)
  connect(fixed1.flange, spring1.flange_a);
  connect(spring1.flange_b, mass1.flange_a);
end Oscillator;
```

The additional equations, which are obtained from component connections, are given after the `equation` keyword. For example the `connect(fixed1.flange, spring1.flange_a)` command connects the flanges of the `fixed1` and `spring1` components and creates the additional system of equations:

```
fixed1.flange.s = spring1.flange_a.s;
fixed1.flange.f = -spring1.flange_a.f
```

The model code can be prepared using any text editor or the Modelica Development Tooling module (Pop et al., 2006) of the Eclipse development environment. Simulation of a model requires the OpenModelica environment (Fritzson et al., 2005). To start calculations and to plot the curve, which describes the position of `mass1` component with time, enter the following into the OpenModelica Shell:

```
loadModel(Modelica)
loadFile("Pycodyn.mo")
simulate(Pycodyn.Oscillator, stopTime=10)
plot(mass1.s)
```

The typical stages of translating and executing a Modelica model are (*Fritzson, 2015*): translation (obtaining a flat set of equations, constants, variables and function definitions from Modelica-code), analysis (equations sorting, convert the coefficient matrix into block lower triangular form), optimization (elimination of most equations, converting equations to assignment statements), code generation (obtaining a C-code), compilation (obtaining an executable) and simulation. However, there are alternative ways of executing Modelica (*Fritzson, 2015*).

## Description of modeling principle in Python

The principle of component-oriented modeling in Python is described below and an example of the implementation of a similar oscillator model is shown.

1. Components are described by Python classes that are structurally similar to Modelica classes and have the following attributes: constant parameters and SymPy symbols (analogs of parameters and variables in Modelica), SymPy symbolic equations (difference or DAE), pins for connecting components into a system (analogs of connectors in Modelica). A dynamic system consisting of components is also described by a Python class which attributes are a list of components and equations (together with additional equations for connecting components).

2. If the dynamic behavior of the components is described by difference equations, then the user must describe these equations in the class by replacing the derivatives with the selected difference scheme (e.g., by the Euler method).

3. The initial conditions are substituted into these difference equations and unknowns are found by solving a system of nonlinear equations at each step or using a function that was initially created using the SymPy `lambdify` function (translates a SymPy expression into an equivalent numeric function) and calculate unknown values at current step from the known values from the previous step without the need to solve the equations.

4. At each step, the `if` statement checks for discrete events that depend on state or time. During event handling, initial conditions, components, or equations can be changed.

5. If the dynamic behavior of the components is described by DAEs, then the Assimulo interface to the DAEs solvers is used, which has an effective discontinuity handling procedure.

6. The SymPy package allows arbitrary manipulation of model equations and code generation. You can solve some algebraic or differential equations symbolically. The DAE system of equations should be simplified, transformed into an ODE, and solved by the SymPy `dsolve` function.

Now we will develop the `pycodyn` module with similar components in Python (Listing S2). The behavior of the components will be described using the difference equations. For simplicity, we will use the Euler method. As a result, the system of components connected by flanges will be described by the system of the difference equations. First, import the `sympy` module and the standard mathematical module `math`. It is important to distinguish the functions of these modules.

```
from sympy import *
import math
```

Create the global variable `dt` (time step).

```
dt=0.1
```

If you only need to obtain the system of equations in a symbolic form, then this variable must be an instance of the `Symbol` class from the `sympy` module:

```
dt=Symbol('dt')
```

`Translational1D` is the basic class of mechanical 1D components with translational motion. The `__init__` method is called when an object of this class is created and has two parameters—the name of the component (`name`) and the dictionary of its attributes (`args`). For component attribute naming, we use the following notation. At the beginning of the name, the `x`, `v`, `a`, `f` symbols mean position, speed, acceleration and force, respectively. At the end of the name, the `p` symbol means the value at time `t-dt`. The numerical index at the end corresponds to the flange number. To distinguish the variables of various components in the system, each of them begins with the name of the component followed by the symbol "`_`". For example, the `s1_x2p` name means the position of the second flange of the `s1` component at time `t-dt`. The `__init__` method for each name-value pair of the dictionary `args` (except `name` and `self`) creates SymPy variables. The symbolic variable of the `Symbol` class is created if its value is not known. In another case, the numeric variable of the `Number` class is created. The `self.eqs` list contains the component equations, and the `self.pins` list contains the component flanges. Each equation is created using SymPy class `Eq`. Each flange is described by a dictionary, which keys are `x`, `xp`, `f`, and the values are the corresponding attributes of the component (see `Mass`, `SpringDamper`, `Force` classes). The `pinEqs` method returns a list of equations for the component flange that is connected to the flanges of the other components. It has the `pindex` parameter—the index of the flange (e.g., 0), and the `pins` parameter—the list of other components flanges. Always the positions of the mechanical 1D translational component on the flange are equal, and the sum of the forces on this flange is zero. For example, if the flange 2 of an `s1` component is connected to the flange 1 of an `m1` component then `pinEqs` method of the `s1` component returns the list of equations [`s1_x2==m1_x1`, `s1_x2p==m1_x1p`, `s1_f2==-m1_f1`].

```
class Translational1D(object):
    def __init__(self, name, args):
        self.name=name # component name
        for k,v in args.items(): # for each key-value pair
            if k in ['name','self']: continue # except name and self
            if v==None: # if value is None
                # create symbolic variable with name name+'_'+k
                self.__dict__[k]=Symbol(name+'_'+k)
            elif type(v) in [float,Float]: # if value is float
```

```
                    self.__dict__[k]=Number(v) # create constant
            self.eqs=[] # equations list
            self.pins=[] # pins list

        def pinEqs(self,pindex,pins):
            eqs=[] # equation list of the flange
            f=Number(0) # sum of forces on flanges of other components
            for pin in pins: # for each flange of the other components
                # add equations describing the equality on the flange:
                # positions
                eqs.append(Eq(self.pins[pindex]['x'], pin['x']))
                # positions at time t-dt
                eqs.append(Eq(self.pins[pindex]['xp'], pin['xp']))
                f+=pin['f'] # add to the sum of forces
            # equality to zero the sum of forces on the flange
            eqs.append(Eq(self.pins[pindex]['f'], -f))
            return eqs
```

The `Mass` class describes the mass concentrated at a point, which has a translational motion. It inherits `Translational1D` class. The `__init__` constructor calls the constructor of the base class `Translational1D` and sends to it the parameters `name` and `locals()`. The latter is a dictionary of local variables `self, name, m, x, xp, v, vp, a, f1, f2`.

```
class Mass(Translational1D):
    def __init__(self, name, m=1.0, x=None, xp=None, v=None, vp=None,
a=None, f1=None, f2=None):
        Translational1D.__init__(self, name, locals()) # base class
constructor call
        # system of equations
        self.eqs=[Eq(self.m*self.a, self.f1+self.f2),
                Eq(self.a, (self.v-self.vp)/dt),
                Eq(self.v, (self.x-self.xp)/dt)]
        self.pins=[dict(x=self.x, xp=self.xp, f=self.f1),
                dict(x=self.x, xp=self.xp, f=self.f2)] # two flanges
```

The behavior of this component is described by a system of equations `self.eqs`. For example, for an `m1` component:

```
[m1_m*m1_a == m1_f1+m1_f2, m1_a == (m1_v- m1_vp)/dt,
m1_v == (m1_x-m1_xp)/dt]
```

A list of additional equations can be generated for each component flange using the `pinEqs` method described above. The first element of the `self.pins` list is the dictionary `dict(x=self.x, xp=self.xp, f=self.f1)`, which means that the x, xp positions on the flange will be equal to the `self.x`, `self.xp` attributes of this component,

respectively, and the force `f` on the flange will be equal to the `self.f1` attribute. The same applies to the second element of the list.

The `SpringDamper` class (Listing S2) describes the translational 1D spring and damper, which are connected in parallel. It inherits `Translational1D` class. In addition to the attributes described above, it has the following attributes: spring constant `c`, damping constant `d`, relative velocity between flanges `vrel`. The behavior of this component is described by a system of equations `self.eqs`. E.g. for an `s1` component:

```
[s1_c*(s1_x2-s1_x1)+ s1_d*s1_vrel == s1_f2, -s1_f2 == s1_f1,
s1_vrel == (s1_x2-s1_x2p)/dt-(s1_x1-s1_x1p)/dt]
```

This component also has two flanges and it is possible to generate a list of additional equations using the `pinEqs` method.

The `Force` class (Listing S2) describes a 1D force with a translational motion of the application point. The value of the `f` force can be constant or variable. It inherits the `Translational1D` class and has one flange.

The `System` class (Listing S2) describes the system of components connected by flanges. The constructor `__init__` gets two parameters—the list of components `els` and the list of additional equations `eqs`, which usually are created using `pinEqs` method. The system components are stored in the `self.els` list and the `self.elsd` dictionary. The list `self.eqs` contains all system equations and is created by joining the equations of all components with additional equations `eqs`.

The `solve` method of this class solves a stationary problem. It returns the solution of a system of equations with conditions `ics`—a dictionary with known values of variables. To solve a system of equations, it can use the SymPy `solve` function, but its algorithm is very slow. It is possible to use fast algorithms for solving equations, for example, the function `scipy.optimize.root` from the SciPy library, which supports many effective methods for solving nonlinear systems of equations. In this case, the call of the SymPy function `solve(eqs)` must be replaced with the call of the `self.solveN(eqs)` method, which adapts the system of equations for SciPy and solves it using `scipy.optimize.root`.

The `solveDyn` method solves a non-stationary problem. It receives three parameters—the dictionary with initial state `state`, the final time value `timeEnd` and the `fnBC` function that returns the dictionary to update the state. First, the time variable `t` is assigned an initial value. In the `while` loop with the condition `t<timeEnd`, the following instructions are executed: previous step variables (`xp`, `x1p`, `vp`, etc.) are assigned the values of the initial state `state`, the values of the boundary conditions are updated, the system of equations is solved by calling the `self.solve` method, solutions are assigned to the dictionary `state`, the results are saved, the time value increases by `dt`. After the loop is completed, the method returns the results as `T` and `Res` lists. These results can be represented in the form of plots using the `matplotlib` library.

But the use of the `self.solve` method can be acceptable only for very frequent discontinuities that require the re-creation of equations. Since at each step this method creates and solves a system of nonlinear equations, the calculations can be very

time-consuming. In most cases, it should be replaced by the `solvN` method, which at each step finds unknown values by passing the values found in the previous step to the `ceqsf` method. At the beginning of the simulation and after the discontinuities, this method must be created using SymPy `lambdify` function that transforms SymPy expressions to lambda functions which can be used to calculate numerical values very fast. This is done in the `createCurEqs` method.

Event processing is performed at the end of each step by calling the user-defined event handling method `event(state)`. In it, the `if` statement checks a determined condition with `state`. If the result is `True`, then the event is handled, for example, new boundary conditions are created and `createCurEqs` is called. You can easily implement modeling of variable structure systems by calling in the `event` method the constructor of the `System` class (with new values of `els`, `eqs`) and the `createCurEqs` method.

If differential equations are used in the components, then the functions and their derivatives should be distinguished. The presence of the symbol "D" in the name of the variable means derivative. For example, `m1_Dx` is a derivative of `m1_x`. Class descriptions of such components will be more like Modelica classes (Table 1). In this case, to solve the DAEs in the form $0 = F(t, y, y')$, the `solveDAE` method from the pycodynDAE (Listing S3) module is used. Assimulo was used as an interface with ODE/DAE solvers such as SUNDIALS IDA (*Hindmarsh et al., 2005*) or DASSL (*Petzold, 1982*). The `solveDAE` method forms the `residual` method and initial values for the time, states and state derivatives required by a DAEs solver. The `residual` method takes as input time $t$, state $y$, state derivative $y'$ and returns a residual vector (zero if a solution is found). Argument lists for the `residual` are prepared by the `residualArgs` method. It is also possible to create user-defined functions `state_events` and `handle_event` for event tracking and handling in discontinuous problems for Assimulo.

Some problems in pycodyn, which is formulated using difference or differential equations, can be solved symbolically using the SymPy functions `solve` and `dsolve`. In particular, the `solve` function, which symbolically solves equations and systems of equations, helps to form the `ceqsf` method mentioned above. The `dsolve` function solves any supported kind of ODEs. Therefore, DAEs needs to be transformed into ODEs by simplification.

## USE CASES

For testing purposes of pycodyn, we consider models of sucker rod strings. Let's take a look at the steel sucker rod string, in which the length is 1,510 m. Such a string and its practical dynamometer card are described in (*Belov, 1960*). The upper section of the string consists of 695 m rods with a diameter of 22 mm, and the lower section consists of 815 m rods with a diameter of 19 mm. This string will have a total mass of 3,961 kg, a total weight in the liquid of 34,687 N, a spring constant of 44,650 N/m, a damping constant of 2,121 N·s/m. Liquid weight above the pump with a diameter of 43 mm will be 18,499 N.

**Table 1 The Mass class in Python (on the left) in comparison with the same class in Modelica (on the right).**

```
class Mass(Translational1D):
    def __init__(self, name, m=1.0, x=None,
v=None, a=None, f1=None, f2=None,
Dx=None, Dv=None):
        Translational1D.__init__(self, name,
locals())

    self.eqs=[
        Eq(self.m*self.a,self.f1+self.f2),
        Eq(self.a, self.Dv),
        Eq(self.v, self.Dx)]

    self.pins=[dict(x=self.x, f=self.f1),
               dict(x=self.x, f=self.f2)]
```

```
model Mass
    extends Transl;
    parameter Real m(start=1);
    Real s;
    Real v(start=0);
    Real a(start=0);

equation
    m*a=flange_a.f+flange_b.f;
    a=der(v);
    v=der(s);

    flange_a.s = s;
    flange_b.s = s;
end Mass;
```

## Use case 1: simulation of free vibrations of the sucker rod string

First, simulate the free vibrations of the string using the Modelica language. Initially, the string is stretched by moving the lower end of the string (`mass1.s`) by one m. After the lower end is released, free vibrations will begin. Use the model (Listing S1) with parameter values: `mass1.m=3961.0`, `spring1.c=44650.0`, `spring1.d=2120.7`, and with initial conditions: `mass1.s=−1`, `mass1.v=0`. Simulation options: `stopTime = 10.0`, `numberOfIntervals = 500`, `tolerance = 1e-006`, `method = 'dassl'`.

Now perform the simulation of free vibrations of the sucker rod string in pycodyn (Fig. 1). In the separate module (Listing S4) create the components: spring-damper `s1` and mass `m1`. In round brackets, there are the values of the attributes—the name and the known parameters.

```
from pycodyn import *
s1=SpringDamper(name='s1', c=44650.0, d=2120.7)
m1=Mass(name='m1', m=3961.0)
```

Create the list of additional equations, formed by connecting the flanges of the components. Then create the object of the component system.

```
peqs=s1.pinEqs(1,[m1.pins[0]])
s=System(els=[s1,m1], eqs=peqs)
```

A list of the model equations can be printed using the command `print(s.eqs)`. To obtain equations only in the symbolic form, the numerical values of the constructor parameters c, d, m should be replaced by None:

```
[s1_c*(-s1_x1 + s1_x2) + s1_d*s1_vrel == s1_f2,
-s1_f2 == s1_f1,
s1_vrel == -(s1_x1 -- s1_x1p)/dt + (s1_x2 -- s1_x2p)/dt,
m1_a*m1_m == m1_f1 + m1_f2,
m1_a == (m1_v -- m1_vp)/dt,
```

```
m1_v == (m1_x -- m1_xp)/dt,
s1_x2 == m1_x, s1_x2p == m1_xp, s1_f2 == -m1_f1]
```

Solve the static problem—the string is stretched by one m.

```
ics={m1.x:-1.0,m1.v:0.0,m1.a:0.0,s1.x1:0.0,s1.x1p:0.0,m1.vp:0.0}
d=s.solve(ics)
```

The boundary conditions depend on the type of problem. If this is the problem of free oscillations, then the position of the string top point `elsd['s1'].x1` and the force on the plunger `elsd['m1'].f2` are zero. Create the function to update the boundary conditions at time `t` for the `fnBC.vrs` components. Then solve the dynamic problem—free vibrations of the string.

```
def fnBC(d, t):
    val = 0.0, 0.0
    return dict(zip(fnBC.vrs, val))
fnBC.vrs = s.elsd['s1'].x1, s.elsd['m1'].f2
T,R=s.solveDyn(d, timeEnd=10, fnBC=fnBC)
```

It is possible to improve the results in the Python model by using the more accurate but more complex difference schemes. For example, if the trapezoidal rule is used (Listing S5), the second and third equations for the `Mass` should be

```
Eq((self.a+self.ap)/2, (self.v-self.vp)/dt),
Eq((self.v+self.vp)/2, (self.x-self.xp)/dt)
```

Consider using components with DAEs and the Assimulo interface from the pycodynDAE module (Listing S3). Create components and a system in the same way and first solve the static problem—the string is stretched by one m (Listing S6).

```
bc={s1.x1:0.0, s1.Dx1:0.0} # constant boundary conditions
eq=s.eqs.subs(bc)
ics={m1.x:-1.0, m1.v:0.0, m1.a:0.0, s1.Dx2:0.0}
state=s.solve(eq, ics)
```

Use the SUNDIALS IDA solver with absolute and relative tolerances 1e-06 to solve the dynamic problem—free vibrations of the string.

```
state.update(ics)
eq=eq.subs({m1.f2:0.0}) # additional BC
T,Y,Yd=s.solveDAE(eq, state, 10.0)
```

Now solve the equations symbolically using the ODE-solver SymPy (Listing S7). Substitute the boundary conditions into the system of equations (`m1.f2 = 0.0`, `s1.x1 = 0.0`, `s1.Dx1 = 0.0`), simplify the system, and after substitution of functions and derivatives instead of symbols, obtain the well-known equations of the harmonic oscillator.

```
Eq(2120.0*m1_v(t) + 44650.0*m1_x(t), -3961.0*Derivative(m1_v(t), t)),
Eq(m1_v(t), Derivative(m1_x(t), t))
```

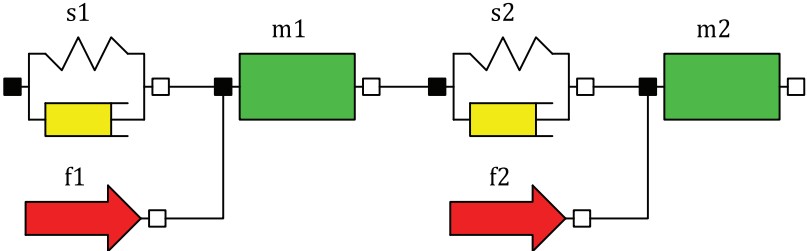

**Figure 2 Component diagram of the model with two sections.**

Solve the equation using `dsolve` with the initial conditions `m1_x(0) = −1.0`, `m1_v(0) = 0.0` and obtain a well-known solution:

```
Eq(m1_v(t), 3.36816*exp(-0.26761*t)*sin(3.34676*t)),
Eq(m1_x(t), -(0.08*sin(3.34676*t) + cos(3.34676*t))*exp(-0.26761*t)
```

## Use case 2: simulation of the pumping process

During pumping, the following loads act on the string: the weight of the rods, the weight of the liquid (only during the upstroke) and dynamic loads (*Belov, 1960*; *Gibbs, 2012*). To build the single-section model `Pumping` in Modelica (Listing S1), use the components of the oscillator and additional components: `motion1` (describes the movement of the upper point) and `force1` (describes the forces acting on the lower point). This is a simplified model that does not take into account other types of loads (*Kopey, Kopey & Kuzmin, 2017*). The stroke length of the upper point is 2.1 m, the number of double strokes per minute is 6.4. During the downstroke, the weight of the rods acts on the lower point. During the upstroke, the weight of the liquid is added to it. This is shown in the algorithm section of the `Pumping` model:

```
algorithm
  if mass1.v <= 0 then
    force1.f:=34687.0;
  else
    force1.f:=34687.0+18499.0*tanh(abs(mass1.v )/0.01);
  end if;
```

The analog of this single-section model in Python is shown in (Listing S8). Now in the new module (Listing S9) create the two-section Python-model of the sucker rod string, in which each section of the string is modeled by three components. The model of each section consists of three 1D mechanical translational components: `SpringDamper`, `Mass` and `Force` (Fig. 2). The `SpringDamper` component is designed to simulate the elastic-damper properties of the string section, the `Mass` component simulates the inertial properties of the section, and the `Force` component simulates the section weight in the fluid and other external forces acting on the section. The upper section has a mass of 2,112 kg, a weight in the liquid of 18,494 N, a spring constant of 114,926 N/m, a damping constant of 5,458 N·s/m. The lower section has a mass of 1,850 kg, a weight in the liquid of 16,193 N, a spring constant

of 73,021 N/m, a damping constant of 3,468 N·s/m. Assign values to the variable of sections weights `fs` and the variable of liquid weight above the plunger `fr`.

```
from pycodyn import *
fs=(-18494.0, -16193.0)
fr=-18499.0
```

Create the components: the spring-damper of the first (upper) section `s1`, the mass of the first section `m1`, the weight of the first section `f1`, the spring-damper of the second section `s2`, the mass of the second section `m2`, the weight of the second section with the weight of the liquid `f2`.

```
s1=SpringDamper(name='s1,' c=114926.0, d=5458.0)
m1=Mass(name='m1',m=2112.0)
f1=Force(name='f1', f=fs[0])
s2=SpringDamper(name='s2', c=73021.0, d=3468.0)
m2=Mass(name='m2', m=1850.0)
f2=Force(name='f2')
```

Create the list of the additional equations (formed by connecting the component flanges) and the object of the component system (the string model).

```
peqs=s1.pinEqs(1,[m1.pins[0]])
peqs+=m1.pinEqs(1,[s2.pins[0],f1.pins[0]])
peqs+=s2.pinEqs(1,[m2.pins[0]])
peqs+=m2.pinEqs(1,[f2.pins[0]])
s=System(els=[s1,m1,s2,m2,f1,f2], eqs=peqs)
```

The complete list of equations for this system `s.eqs` in the SymPy format:

```
[s1_c*(-s1_x1 + s1_x2) + s1_d*s1_vrel == s1_f2,
-s1_f2 == s1_f1,
s1_vrel == -(s1_x1 - s1_x1p)/dt + (s1_x2 - s1_x2p)/dt,
m1_a*m1_m == m1_f1 + m1_f2,
m1_a == (m1_v - m1_vp)/dt,
m1_v == (m1_x - m1_xp)/dt,
s2_c*(-s2_x1 + s2_x2) + s2_d*s2_vrel == s2_f2,
-s2_f2 == s2_f1,
s2_vrel == -(s2_x1 - s2_x1p)/dt + (s2_x2 - s2_x2p)/dt,
m2_a*m2_m == m2_f1 + m2_f2,
m2_a == (m2_v - m2_vp)/dt,
m2_v == (m2_x - m2_xp)/dt,
s1_x2 == m1_x, s1_x2p == m1_xp,
s1_f2 == -m1_f1, m1_x == s2_x1,
m1_xp == s2_x1p, m1_x == f1_x,
m1_xp == f1_xp, m1_f2 == f1_f - s2_f1,
s2_x2 == m2_x, s2_x2p == m2_xp,
```

```
s2_f2 == -m2_f1, m2_x == f2_x,
m2_xp == f2_xp, m2_f2 == f2_f]
```

Solve the static problem—the string under the maximum static loads.

```
ics={m1.v:0.0, m1.a:0.0, m2.v:0.0, m2.a:0.0, s1.x1:0.0, s1.x1p:0.0,
f2.f:fs[1]+fr}
d=s.solve(ics)
```

Dictionary `d` contains the results. To display the position value for the bottom point of the second section, enter the command `print(d[m2.x])`. We get the result −0.94. This is the elongation value of the string under the maximum load.

Solve the dynamic problem—the upper point has a harmonic motion. The `motion` function describes the harmonic motion of the upper point and returns its position at time `t`.

```
def motion(t):
    A=2.1/2 # amplitude
    n=6.4/60 # frequency
    return A*math.sin(2*math.pi*n*t) # position
```

The `force` function returns the value of the force on the pump plunger `F`, depending on the value of its speed `v`. If the speed is less than zero (downstroke of the string), the function returns the weight value of the second section. Otherwise, the function returns the sum of the second section weight and the liquid weight above the plunger. This function should be smoothed when the sign of the velocity changes, for example, using the `math.tanh` hyperbolic tangent function.

```
def force(v):
    F=fs[1] # weight of the second section
    if v>0: # if upstroke
        F+=fr # increase the force by value of the fluid weight
    return F*math.tanh(abs(v)/0.01) # smoothing near the point v=0
```

Create the function to update the boundary conditions at time `t` for `fnBC.vrs` components. Here, `d` is the dictionary of the results calculated in the previous step. Then solve the problem.

```
def fnBC(d, t):
    val = motion(t), force(d[m2.v])
    return dict(zip(fnBC.vrs, val))
fnBC.vrs = s.elsd['s1'].x1, s.elsd['f2'].f
T,R=s.solveDyn(d, timeEnd=20.0, fnBC=fnBC)
```

The simulation of the variable structure system (breakage of the second section) by the Euler method (`dt=0.1`) is implemented in (Listing S10). If the force at the top of the second section is greater than 56,000 N, then the section breaks off. The user method `event` is created to handle the event. At each step, this method checks the condition `state[s1.f1]>56000`. If the result is `True`, then an event occurs. The handling of this

event consists in changing the components of the system (only `s1`, `m1`, `f1` remain after the breakage), changing additional equations at the connection points, changing the boundary conditions `fnBC` (the weight of the second section and the weight of the liquid are zero). The `ceqsf` method is also updated by calling `createCurEqs`.

```
def event(self, state):
    if state[s1.f1]>56000:
        self.fnBC=fnBC2
        peqs=s1.pinEqs(1,[m1.pins[0]])
        peqs+=m1.pinEqs(1,[f1.pins[0]])
        self.__init__(els=[s1,m1,f1], eqs=peqs) # changing the system
        self.createCurEqs(fnBC2) # new current equations
System.event=event
```

Analogous single-section (Listing S11) and two-section (Listing S12) models, based on differential equations, are developed using the pycodynDAE module. The DAE system should be supplemented with equations that describe the position of the upper point and the force that acts on the lower point. For example:

```
eq=s.eqs+Tuple(Eq(s1.x1, A*sin(2*pi*n*t)),
                Eq(m1.f2, Piecewise((fs, m1.v<0),
                    (fs+fr*tanh(abs(m1.v)/0.01), m1.v>=0)))))
```

Alternatively, you can replace the symbols `s1.x1` and `m1.f2` with the right-hand sides of these equations.

## SIMULATION RESULTS

The results of the simulation of free vibrations are shown in Fig. 3. The frequency of free vibrations corresponds to the theoretical natural frequency of the harmonic oscillator $\omega = (j/m)^{1/2} = (44{,}650/3{,}961)^{1/2} = 3.357$ rad/s, where $j$ is the stiffness, $m$ is the mass. Such vibrations occur during normal operation of the pump due to the sharp removal or application of the load (the pump valve opens or closes) and are noticeable in the upper and lower parts of the dynamometer card (*Belov, 1960*). The differences are explained by the use of unequal difference schemes. The results obtained analytically and by IDA/DASSL solvers are almost equal, therefore they are shown by a single curve in the figure. The global error values (at `t=1` s) for the Euler (`dt=0.1`), trapezoidal rule (`dt=0.1`), Euler (`dt=0.01`), DASSL (OpenModelica), SUNDIALS IDA methods are, respectively, 0.344, 0.034, 0.046, 6.35E-05, 1.46E-05. The simulation time is 0.25, 0.5, 2.46, 0.29, 0.028 s, respectively. The total time of the module execution (total simulation time for OpenModelica) is 4.1, 5.5, 6.3, 5.1, 1.9 s, respectively. These data were obtained for this configuration: simulation interval 0–10 s, CPU 2.5 GHz, Python 3.7, NumPy 1.16.4, OpenModelica 1.12, Sundials 2.6.

The results of the pumping process simulation (Fig. 4) correspond to practical dynamometer cards obtained on real wells (*Belov, 1960*). Single-section (Listing S8) and two-section (Listing S9) models, simulated by the Euler method (d$t$ = 0.1), somewhat

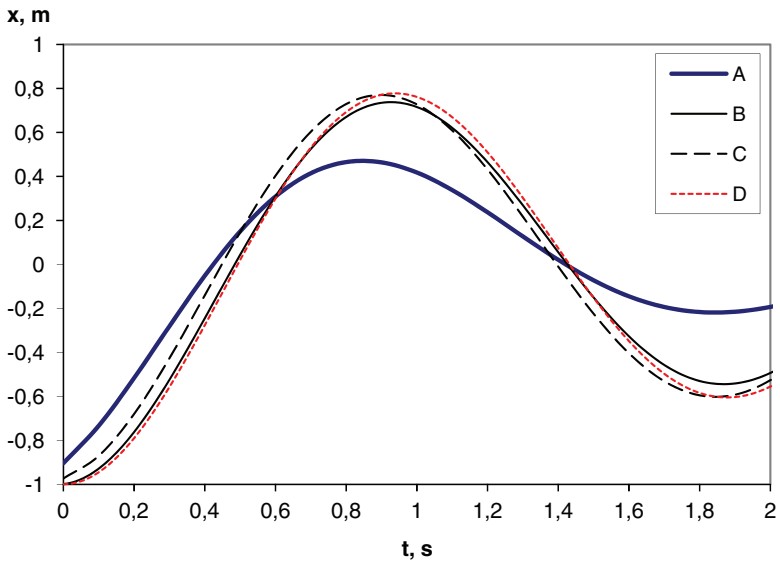

**Figure 3 Plunger position (x) during free oscillation of the string.** (A) Euler method with time step $dt = 0.1$ s; (B) Euler method with time step $dt = 0.01$ s; (C) Trapezoidal rule with time step $dt = 0.1$ s; (D) DASSL (Modelica-model), SUNDIALS IDA, analytical solution.

distort the right and left sides of the card and smooth the upper and lower sides (Fig. 4A). Single-section (Listing S11) and two-section (Listing S12) models, simulated by the IDA, give somewhat larger values of maximum load and lower values of minimum load (Fig. 4B). The two-section model is more adequate.

The dynamometer card of the string breakage model (Fig. 5) corresponds to practical dynamometer cards with their typical flat shapes (*Belov, 1960*).

## DISCUSSION

It is noticeable that the global error for the Euler method (`dt` = 0.1) is much larger, therefore it is not advisable to use it for the free vibration problem. In addition, numerical solution procedure in pycodyn has a low performance if difference equations are used. In the future, it is planned to improve performance, for example by using Cython.

The differences with the practical dynamometer card are explained by the fact that the real string has a greater number of degrees of freedom. For a more adequate simulation, you need to increase the number of sections of the model (*Kopey, Kopey & Kuzmin, 2017*) or use the wave equation, which is a partial differential equation (*Gibbs, 2012*). In addition, it is difficult to know the exact value of the damping constant, which depends on many factors (*Kopey, Kopey & Kuzmin, 2017*). In general, all these models can be used for approximate modeling of the pumping process.

The Python language allows a simple modification and improvement of pycodyn. In the future, it is planned to extend the set of the components (e.g., create electrical and hydraulic components), develop support for hierarchical subsystems and the tools for building models using component diagrams. To implement hierarchical subsystems, you can move functions for solving equations to a separate module and add pins to

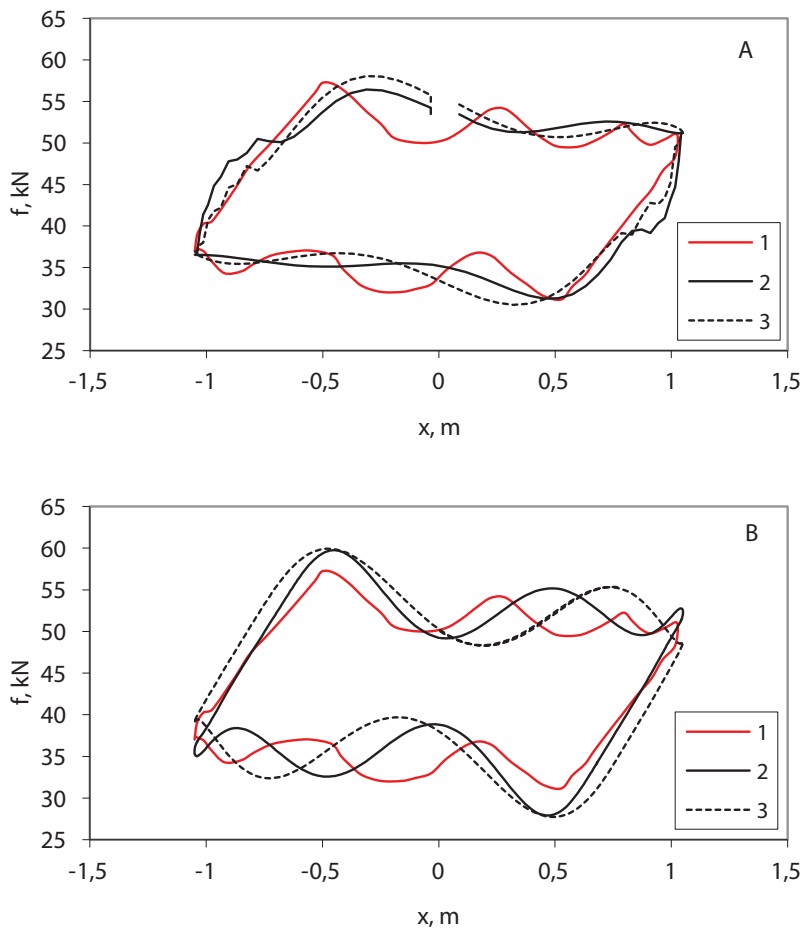

**Figure 4 Simulation results—wellhead dynamometer cards.** (1) Practical dynamometer card; (2) two-section model; (3) single-section model; (A) pycodyn with Euler method, d$t$ = 0.1; (B) pycodynDAE with SUNDIALS IDA.               

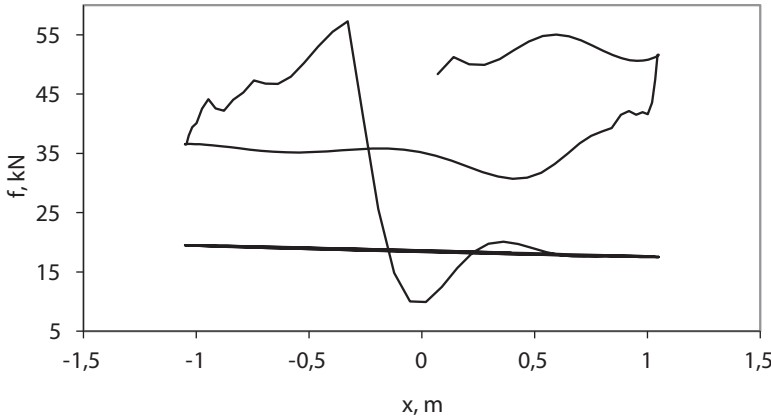

**Figure 5 The simulation of the breakage of the sucker rod string (wellhead dynamometer card).**

the `System` class. The problem in the form of difference equations is usually more difficult to formulate. But SymPy can be used to automate the conversion of differential equations to difference equations. In order to simplify the code of the pycodyn and pycodynDAE modules, the algorithms for equations sorting, eliminating, and simplifying are not implemented. This is planned to be implemented in the future using SymPy.

## CONCLUSIONS

The Python-classes that allow creating the Modelica-like models in Python without the need to study and apply specialized modeling languages are developed. The suggested approach simplifies the understanding of the system, its modification and improvement, adaptation for other purposes, makes it available to a much larger community, simplifies integration into third-party software.

Difference or differential equations can be used to describe components. Using difference equations and the pycodyn module allows simplifying the implementation of the hybrid modeling, variable structure systems modeling and the requirements for the modules for symbolic mathematics and for solving equations. It is also well suited for experimenting with evolutions of modeling capabilities. In particular, one can experiment with arbitrary difference schemes, make arbitrary symbolic manipulations and modify numerical solution procedures. However, the pycodynDAE module can provide higher accuracy and performance using third-party DAEs solvers that are suitable for stiff problems. With SymPy, some tasks can be solved symbolically.

The comparison of simulation results of sucker rod string with practical dynamometer cards and Modelica models verify the adequacy of the models. The pycodyn framework can be used to study the principles of component-oriented modeling and for various kinds of experiments on its new features. The source code is freely available under the GNU GPLv3 open-source license from the GitHub (https://github.com/vkopey/pycodyn).

### Funding
The authors received no funding for this work.

### Competing Interests
The authors declare that they have no competing interests.

### Author Contributions
- Volodymyr B. Kopei conceived and designed the experiments, performed the computation work, authored or reviewed drafts of the paper, approved the final draft.
- Oleh R. Onysko performed the experiments, analyzed the data, prepared figures and/or tables, approved the final draft.
- Vitalii G. Panchuk performed the experiments, contributed reagents/materials/analysis tools, authored or reviewed drafts of the paper, approved the final draft.

## Data Availability

Data is available at GitHub: https://github.com/vkopey/pycodyn.

## Supplemental Information

Supplemental information for this article can be found online at http://dx.doi.org/10.7717/peerj-cs.227#supplemental-information.

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
