# Peer review of "Component-oriented acausal modeling of the dynamical systems in Python language on the example of the model of the sucker rod string"

_PeerJ Computer Science, doi:10.7717/peerj-cs.227_

## Round 0.1 · original submission · Major Revisions

Coupled the reviewers’ comments and my own understanding, I suggest the author improve this manuscript by specifically addressing the following comments:

1) Please clearly and simply point out the main ideas and novelty of the manuscript, especially in Abstract and Introduction.

2) Please improve the English language throughout the manuscript.

3) Please also improve the structure of the manuscript according to the comments.

4) Please well prepare the source code as the supplementary.

Reviewer 1 ·

Basic reporting

The article presents a simple component-oriented acausal modelling system for application to a particular type of problems (modelling of a sucker rod string). The system is implemented in Python and utilises the available Python numerical libraries such s SciPy and SymPy. Models are specified using Python classes describing reusable components. Model variables are represented by SymPy symbols and equations are specified in a symbolic form using SymPy package. Differential equations are internally discretised in time using the first order Euler and trapezoidal rule methods yielding a system of non-linear equations (“difference equations”). Model instances are connected using pins producing an additional set of connectivity equations. Equations from the hierarchical model definition are collected and numerically solved using SymPy or SciPy packages. The system is applied to modelling of a sucker rod string and results compared to the results from Modelica.

The article requires significant changes and major revisions. In the next revision the following comments should be addressed by the authors.

The English language must be thoroughly improved throughout the manuscript. In its current state the manuscript is not acceptable for publication.

Introduction and the background are described well and the prior work is well referenced (again, the English language should be improved).

The article is not well structured and I suggest the following changes (in accordance with an acceptable format of ‘standard sections’):
1. Introduction (including the problem statement)
2. Methods/Implementation (the main characteristics and principles, and the implementation details of the pycodyn module, description of the components etc.)
3. Use cases/applications
3.1 Case 1: Simulation of free vibrations of the sucker rod string (full description of the problem, including equations for every component and references in the literature regarding the theoretical background)
- Model implementation(s) in Python
- Model implementation in Modelica
3.2 Case 2: Simulation of the pumping process by the two-section string (full description of the problem, including equations for every component and references in the literature regarding the theoretical background)
4. Results
5. Discussion
6. Conclusions
This is only a general idea, the authors are free to modify it.

The Results and Discussion sections are missing and the simulation results are only briefly presented but not discussed at all. The focus in the Implementation section should be more on general principles of the modelling system rather than explanation of the Python code.

Source code listings are given in both manuscripts and as as supplemental information. I suggest to keep the supplemental listings and remove from the manuscript those starting at lines 283, 327, 352, 368 and 398. The parts of interest can be referenced in the main text by a class or a function name.
Supplemental listings should be reorganised in the following way:
- Supplemental listing S1 should contain the pycodyn module(s) for two methods used (Euler and trapezoidal rule)
- Supplemental listing S2 should contain the Python model from Case 1
- Supplemental listing S3 should contain the Modelica model from Case 1
- Supplemental listing S4 should contain the Python model from Case 2
Again, this is only a general idea, the authors are free to modify it.

The provided source code for Case 1 and 2 models should be prepared for running in Python with no modifications from readers. In addition, the current models work only in Python 2 and should be modified to work in Python 3, too.

Experimental design

The research falls within the aims and scope of PeerJ Computer Science. However, the main idea and the novelty of the framework are not clearly described. A comparison to the available modelling approaches should be given and discussed in details (not only listed).

I encourage the idea of modelling frameworks based on high level programming languages and reuse of existing libraries for numerical integration. However, in general, performing a simulation requires the following tasks: (1) model specification (simulator-dependent), (2) internal processing of the model specification and generation of a system of equations, and (3) numerical solution of a system of equations (in general, simulator-independent). While the idea of basing a component-oriented system for model specification on SymPy is plausible, the numerical solution procedure in this work is not satisfactory and requires significant improvements. For instance, the numerical performance of the current implementation is extremely low and application to stiff problems is problematic. Furthermore, the numerical stability and accuracy are not taken into consideration (important for the first order Euler method) and the local and the global error control are not performed.

On the one hand, description of models and generation of a system of equations can be performed in many different ways depending on the type of the problem and the method applied by a simulator. On the other hand, the numerical solution procedure always requires the same information and is rather generic. Hence, the existing simulators provide either a modelling language or an API for model specification but perform the numerical solution process using the state-of-the-art ODE/DAE and linear solvers implemented in C/C++ (for performance reasons). Very low numerical performance in the pycodyn system can be resolved by using some of the available DAE solvers. For instance, the SymPy symbols for time derivatives can be added to components in the same way as for the variables. Furthermore, by assigning an index to every variable (and corresponding time derivative) it could be possible to map SymPy symbols to one-dimensional arrays of floating point values used by a DAE solver. This way it would not be difficult to implement residual and jacobian routines required by a DAE solver (i.e. Assimulo). Thus, the additional code required to use i.e. Assimulo as a DAE solver includes a function to copy the data to and from the solver specified arrays using the mapping between pycodyn symbols and data arrays in the residual and jacobian functions. This way the performance and the quality of the numerical solution will be significantly improved. This part should be straightforward to implement and would provide a valuable addition to the framework.

Validity of the findings

Event generation is listed as a feature in the Abstract and Statement of the problem sections but not implemented.

It is stated that the results correspond to practical dynamometar cards obtained on real wells (line 593). This statement should be supported by providing the data from the real wells and a discussion added to the Discussion section.

·

Basic reporting

The article use clear English and is understandable.

However, first sentence in abstract is too strong and disputable. E.g. "poor interoperability" in the case of Modelica is not true - see FMI standard, various free fmi libraries for python (PyFMI), C++, .NET. "High cost of learning" is disputable as for specific domain experts it's more understandable than general programming language "Complexity" of implementation is disputable again. While libraries might seem complex, implementing models in some domain using existing libraries enables user to express much more complex systems very easily and plausibly: https://doi.org/10.1109/EMBC.2015.7319192 or https://doi.org/10.1016/j.bbe.2017.08.001

Experimental design

Listing from from row 131 (in Modelica) could be compared with the similar parts from row 247 (in Python) as it could be visible some bioler-plate code and major differences - e.g. rather as figure with two columns - one in Modelica and second with corresponding part in Python.

Validity of the findings

In conclusion row 614 "block diagram" is used usually in terms of causal modeling while we're talking about acausal which is opposite of "block diagram" thus I suggest either use "acausal diagram" or "component diagram". The same for caption of figure 1 and 3.

While the difference method is used, it should be discussed more about stability and robustness in conclusion section, it is just mentioned in Introduction. And compare with e.g. DASSL available in Modelica tools.

Additional comments

Congratulation for a very interesting and elegant library allowing to do acausal modeling in Python.
The real strength of the existing specialized modeling language are domain specific libraries, it would be interesting whether such libraries e.g. Modelica libraries can be transpiled to such tools like yours.

I found minor issue in supplementary materials.
The Modelica model is not reproducible in OpenModelica Editor tool. However with small edit is working. It should be wrapped in some package definition e.g.:
'''
within ;
package Pycodyn
extends Modelica.Icons.Package;

[ content of supplementary material.mo]

end Pycodyn;
'''

I suggest to consider to upgrade to Python version 3 (e.g. using 2to3). Python 2 is going to end of life in 2020.

---

## Round 0.2 · Minor Revisions

I am glad to see that all comments have been addressed. However, before accepting this manuscript, I request the authors to make the following minor revisions.

(1) Improve Figure 1 and Figure 2 by adjusting the fonts, and using the software Micosoft Visio, or Adobe Illustrator.

(2) Improve Figure 3 , Figure 4 and Figure 5 using the software Origin.

---

## Round 0.3 · accepted · Accept

The manuscript has been well improved according to the reviewers' and editor's comments. But I still suggest the authors improve the quality of Figure 1, Figure 2, using Adobe Illustrator or Microsoft Visio.